# Marine aerosol properties over the Southern Ocean in relation to the wintertime meteorological conditions

Manu Anna Thomas[1], Abhay Devasthale[1], and Michael Kahnert[1,2]

[1]Swedish Meteorological and Hydrological Institute, Folkborgsvägen 17, 60176 Norrköping, Sweden.
[2]Chalmers University of Technology, Chalmersplatsen 4, 41296 Gothenburg, Sweden.

**Correspondence:** Manu Anna Thomas (manu.thomas@smhi.se)

**Abstract.**

Given the vast expanse of oceans on our planet, marine aerosols, and sea salt in particular, play an important role in the climate system via multitude of direct and indirect effects. The efficacy of their net impact however depends strongly on the local meteorological conditions that influence their physical, optical and chemical properties. Understanding the coupling between aerosol properties and meteorological conditions is therefore important. It has been historically difficult to statistically quantify this coupling over larger oceanic areas due to the lack of suitable observations, leading to large uncertainties in the representation of aerosol processes in climate models. Perhaps no other region shows higher uncertainties in the representation of marine aerosols and their effects than the Southern Ocean. During winter the Southern Ocean boundary layer is dominated by sea salt emissions.

Here, using 10 years of austral winter period (June, July and August, 2007-2016) space based aerosol profiling by CALIOP-CALIPSO in combination with meteorological reanalysis data, we investigated the sensitivity of marine aerosol properties over the Southern Ocean (40S-65S) to various meteorological parameters, such as vertical relative humidity (RH), surface wind speed, and sea surface temperature (SST) in terms of joint histograms. The sensitivity study is done for the climatological conditions as well as for the enhanced cyclonic and anticyclonic conditions in order to understand the impact of large-scale atmospheric circulation on the aerosol properties.

We find a clear demarcation in the 532 nm aerosol backscatter and extinction at RH around 60%, irrespective of the state of the atmosphere. The backscatter and extinction increase at higher relative humidity as a function of surface wind speed. This is mainly because of the water uptake by the wind driven sea salt aerosols at high RH near the ocean surface resulting in an increase in size, which is confirmed by the decreased depolarization for the wet aerosols. An increase in aerosol backscatter and extinction is observed during the anticyclonic conditions compared to cyclonic conditions for the higher wind speeds and relative humidity, mainly due to aerosols being confined to the boundary layer and their proximity to the ocean surface facilitates the growth of the particles. We further find a very weak dependency of aerosol backscatter on SSTs at lower wind speeds. But when the winds are stronger than about 12 m/s, the backscattering coefficient generally increases with SST.

When aerosol properties are investigated in terms of aerosol verticality and in relation to meteorological parameters, it is seen that the aerosol backscatter values in the free troposphere (pressure < 850 hPa) are much lower than in the boundary layer, irrespective of the RH and the three weather states. This indicates that the local emissions from the ocean surface make the

dominant contribution to aerosol loads over the Southern Ocean. A clear separation of particulate depolarization is observed in the free and lower troposphere, more prominent in the climatological mean and the cyclonic states. For RH > 60%, low depolarization values are noticeable in the lower troposphere, which is an indication of the dominance of water-coated, mostly spherical sea salt particles. For RH < 60%, there are instances when the aerosol depolarization increases in the boundary layer, more prominently in the mean and anticyclonic cases which can be associated with the presence of drier aerosol particles.

Based on the joint histograms investigated here, we provide third degree polynomials to obtain aerosol extinction and backscatter as a function of wind speed and relative humidity. Additionally, backscattering coefficient is also expressed jointly in terms of wind speed and sea surface temperature. Furthermore, depolarization is expressed as a function of relative humidity. These fitting functions would be useful to test and improve the parameterizations of sea salt aerosols in the climate models.

We also note some limitations of our study. For example, interpreting the verticality of aerosol properties (especially depolarization) in relation to the meteorological conditions in the free and upper troposphere (pressure < 850 hPa) was challenging. Furthermore, we do not see any direct evidence of sudden crystallization (efflorescence), deliquescence, or hysteresis effects of the aerosols. Observing such effects will likely require a targeted investigation of individual cases considering tracer transport, rather than the statistical sensitivity study that entails temporally and geographically averaged large data sets.

# 1   Introduction

Aerosols, irrespective of their phase, are defined as particles, suspended in the air with lifetimes ranging from hours to decades. Atmospheric aerosols, both natural and anthropogenic, present themselves in multitude of sizes, shapes and chemical composition. Oceanic sea salt and biogenic emissions, desert dust and volcanic emissions constitute natural sources of aerosols. All others that are as a result of human influence, directly or indirectly such as industrialization, agriculture and forest fires are classified as aerosols derived from anthropogenic sources. Aerosols are very complex in nature and influence the Earth system through their impact on climate, environment and health. Aerosol research focusing on the nature of aerosols and their interactions has progressed significantly over the last decade, given their implications for the Earth's radiative balance. A recent study reports the global mean total effective aerosol radiative forcing to be in the range of -2.0 to -0.35 W/m$^2$ when constrained by observations, with estimates of aerosol-cloud interactions in the range of -2.65 to -0.07 W/m$^2$ (Bellouin et al., 2020).

Among the naturally occurring aerosols, oceans are the dominant source of such aerosols in the form of sea salt with total emission ranging from 1,400 to 6,800 Tg/yr according to the IPCC Fifth Assessment Report (2013). Sea salt aerosols are highly hygroscopic in nature and play an important role in the chemistry of the marine atmosphere (Weng et al., 2020). As they are an important source of halogens, they are capable of altering the chemistry of several species such as ozone, nitrogen, bromine etc via heterogeneous reactions (Sievering et al., 1992; Vogt et al., 1999; Zhu et al., 2019). The dependency of the hygroscopicity of sea salt aerosols on the ambient humidity means that they can impact the amount of radiation that is attenuated thereby affecting the clear sky radiative fluxes (Haywood et al., 1999; Ma et al., 2008). Also, they are very efficient CCN thereby altering the cloud albedo, lifetime and eventually precipitation (Kaufman et al., 2001; Shinozuka et al., 2004; Pierce and Adams, 2006).

Pure sea salt contains 85% salt. Mostly it consists primarily of NaCl and a mixture of one or more of other salts, such as, Mg, K, Ca sulfates etc and traces of organic material. Since sea salt is highly hygroscopic, the ambient relative humidity (RH) determines their morphology and complex refractive index (Tang, 1997; Irshad et al., 2009; Cotterell et al., 2017). In marine environments, RH can reach as high as 80-90%. Deliquescence (the process of water absorption by sea salt particles until they are completely dissolved) and efflorescence (the process in which hydrated salts become drier and return to their crystalline structure) phase transformations of sea salt are more complex and depend on the RH values and the composition. As the RH increases, deliquescence occurs at about 76% RH which is termed as deliquescence RH (DRH). However, as RH decreases, the moist particle does not crystallize at DRH, but at a much lower RH (values below 45%) (Wise et al., 2009). This hysteresis effect has been studied in detail in the laboratory for different salts (Tang, 1996, 1997). Hygroscopic growth of inorganic sea salt is 8-15% lower than that of NaCl (Zeng et al., 2013; Zieger et al., 2017). Sea salt particles come in a range of sizes with dry sea salt aerosols being relatively smaller than those at high RH (Lewis and Schwartz, 2004; Zhang et al., 2005). The influence of RH on aerosol light scattering is evaluated by the scattering enhancement factor which varies considerably depending on the size and type of aerosol and their transport pathway (Zieger et al., 2013). The cloud radar at the Barbados Cloud Observatory which is situated at the windward coast of Barbados (13.16°N, 59.43°W) is found to be well-suited as the propagation of the ITCZ to the SH during NH winter favors the transport of dust-laden air masses to South America leaving this station solely under marine influence (Stevens et al., 2016). One other location that is devoid of anthropogenic influence is the southern Indian Ocean depending on the position of the Mascarene anticyclone (Mallet et al., 2018). These are ideal conditions to explore the sea salt aerosols in detail in their natural environment. Haarig et al. (2017) studied the linear depolarization ratios over Barbados and observed that there were periods with RH below 40% for which the depolarization ratios significantly increased to $0.15 \pm 0.03$ (at 532 nm) compared to the laboratory measurement values of $0.08 \pm 0.01$ for sea-salt crystals (Sakai et al., 2010). This implies that the naturally occurring sea salt particles are not completely dry, but are starting to crystallize. This can have an impact on the Earth's radiative budget.

Apart from the laboratory studies and in-situ observations, numerous campaigns have been carried out to assess the size, morphology and the composition of naturally occurring sea salt (For example: ATom (Atmospheric Tomography mission) aircraft campaign (Froyd et al., 2019; Murphy et al., 2019); The ACE1 (First Aerosol Characterization Experiment) flight campaigns (Bates et al., 1998a); PEMT-B (Pacific Exploratory Mission in the tropics) flight campaign (Fuelberg et al., 2001) etc). All these campaigns were aimed at understanding the physical, chemical and optical properties of tropospheric marine aerosols under clean atmospheric conditions and their dependency on different synoptic states. It was observed that 26% of the accumulation mode particles below 200 m was sea salt and this fraction increases with wind speed by 11% for wind speeds less than 4 m/s and respectively, 20% and 30% at 4-8 m/s and 8-12 m/s. The scattering coefficients of sea salt decreases to 87% under dry conditions compared to 96% at high relative humidities and sea salt contributes to about 63% of the total column AOD (aerosol optical depth) over the Southern Oceans (Shinozuka et al., 2004).

Being one of the dominant aerosol types in the atmosphere and the impact they can instil on the Earth's radiative balance, the role of the sea salt aerosols in the climate system can not be ignored. At the same time, due to their complex nature and high computational costs, the atmospheric flux of sea salt in the general circulation models is heavily parameterized based

on existing campaign measurements. There are two main ways by which sea salt is released into the atmosphere; one by the bursting of air bubbles during whitecap formation and other by the direct tearing of droplets from the top of breaking waves. These two processes release sea salt droplets of varying radii with smaller droplets of size range 0.25 – 8 $\mu$m by the former mechanism and much larger droplets (10- 20 $\mu$m) by the latter (Guelle et al., 2001; de Leeuw et al., 2000). The generation of sea spray in the models are described by source functions that are factors of surface wind speed and/or water temperature and are based on field and/or laboratory data (e.g. Smith et al. (1993); O'Dowd et al. (1997); Andreas (1998); Smith et al. (1993); Vignati et al. (2001); de Leeuw et al. (2000); Monahan et al. (1986); Martensson et al. (2003); Gong (2003)). Clarke et al. (2006) considered ultrafine sea salt as they can have an impact on the CCN concentrations. However, a detailed review of these source functions revealed that the sea spray fluxes varied by several orders of magnitude for different size ranges and wind speeds (Guelle et al., 2001; Vignati et al., 2001; Gong, 2003; Lewis and Schwartz, 2004). All these and many more of these parameterizations were evaluated extensively in the general circulation models (Tsyro et al., 2011; Spada et al., 2015; Neumann et al., 2016; Revell et al., 2019; Weng et al., 2020). It was seen that the atmospheric sea salt concentrations were overestimated by 8 - 46% and the concentrations in precipitation were systematically underestimated by nearly 70% over Europe and can be attributed to the sensitivity to the meteorology used and also uncertainties in the processes such as deposition velocities, in-cloud scavenging ratio etc (Tsyro et al., 2011). Jaeglé et al. (2011) obtained model bias reductions of nearly a factor of two for both cruise and station observations globally when an empirical source function depending on both wind speed and sea surface temperature was used. A recent work by Grythe et al. (2014) suggested a new source function with an averaged 10m wind speed ($u_{10}$) dependency of $u_{10}^{3.5}$ and temperature produced sea salt spray emissions close to the observations. Many of these studies failed to accurately reproduce the aerosol optical depth. For example, the ACCMIP ( Atmospheric Chemistry and Climate Model Intercomparison Project) models and other studies overestimated the winter AOD over the Southern Oceans because of the overestimation of the wind speed dependency introduced by the sea spray source function used in the model (Jaeglé et al., 2011; Shindell et al., 2013; Spada et al., 2015; Revell et al., 2019).

Given their dominance in the atmosphere, multitude of climate impacts, their sensitivity to the local meteorology and their poor representation in global climate models, it is important to continue to study and characterize sea-salt aerosols under varying meteorological conditions. The oceanic area covers approximately 70% of the Earth's surface and since the majority of these water bodies are bounded by continents, it makes it difficult to separate the naturally occurring sea salt aerosols from other natural/anthropogenic aerosols in order to study the characteristic physical and chemical properties in its natural habitat. However, one such area where the anthropogenic influence is relatively negligible is over the Southern Ocean. Using a combination of data from the active aerosol lidar in space and reanalysis, this study therefore investigates aerosol properties over the Southern Ocean in relation to the local meteorology. We chose the SH winter period over the Southern Ocean when solar radiation is at its minimum and DMS emissions are negligible. This is also to make sure that we are predominantly dealing with sea salt aerosols without contamination via long range transport or local sources. With the advancements in active lidar remote sensing, it is now feasible to study the optical properties of sea salt aerosols in much detail in their natural environment over larger spatial scales.

In particular, the answers to the following questions are sought.

1. How sensitive are the backscatter, extinction and depolarization ratio to the surface wind speed, relative humidity and the sea surface temperatures?

2. How is the vertical distribution of aerosol properties impacted by the relative humidity?

3. Do the aerosol properties differ during the cyclonic and anticyclonic conditions?

This study would help understand how deliquescent sea salt aerosols are under varying humidities and how the optical properties of the sea salt aerosols differ at the different stages of the growth.

## 2 Data and methods

### 2.1 Optical properties of aerosol particles

In this study, we make use of the vertical profiles of the aerosol properties such as extinction and backscattering coefficients and linear depolarization ratio, all retrieved from the CALIOP sensor data (flying onboard CALIPSO satellite). The extinction and backscattering coefficients can be used to obtain information on the vertical distribution of the aerosols. The extinction coefficient $k_{\text{ext}} = \mathrm{d}I/(I\mathrm{d}s)$ expresses the relative attenuation of the spectral intensity $I$ of a beam of light along a path of length $\mathrm{d}s$ (Thomas and Stamnes, 2002). In the simplest case of an ensemble of spheres characterized by their radii $r$ and extinction cross sections $C_{\text{ext}}(r)$, it can be computed according to (Sun et al., 2020)

$$k_{\text{ext}} = \int n(r)C_{\text{ext}}(r)\mathrm{d}r, \tag{1}$$

where $n(r)\mathrm{d}r$ denotes the number volume-density of particles in the radius interval $[r, r+\mathrm{d}r]$. Thus $k_{\text{ext}}$ contains information on both the number density and the size distribution of the particles. In the more general case of an ensemble of nonspherical particles of varying microphysical properties, Eq. (1) has to be generalized to an integral over particle sizes, orientations, shapes, and compositions. The backscattering coefficient $k_{\text{bak}}$ is defined analogously, where the extinction cross section $C_{\text{ext}}$ is replaced by the backscattering cross section $C_{\text{bak}}$.

Extinction is the combined effect of absorption and scattering, i.e., $k_{\text{ext}} = k_{\text{abs}} + k_{\text{sca}}$. Marine aerosols are only weakly absorbing, so that $k_{\text{ext}}$ is typically dominated by $k_{\text{sca}}$. Note that the total scattering coefficient $k_{\text{sca}}$ accounts for attenuation by scattering into *all* directions. Thus $k_{\text{sca}}$ is an integral radiative property, while $k_{\text{bak}}$ is a differential radiative property that only depends on the radiative energy scattered into the exact backscattering direction.

The linear depolarization ratio $\delta_l = I_\perp/I_\parallel$ expresses the ratio of perpendicular to parallel polarized intensities of scattered light. In lidar remote sensing, these intensities are measured in the backscatter direction, and the detected intensity $I_\parallel$ is taken to be parallel to the polarization of the emitted laser beam. For an ensemble of particles the linear backscattering depolarization ratio can be modelled according to (Mishchenko et al., 2006)

$$\delta_l^p = \left.\frac{F_{11} - F_{22}}{F_{11} - F_{22}}\right|_{\Theta=180°}, \tag{2}$$

where $F_{ij}$ denote the elements of the $4 \times 4$ Mueller matrix (averaged over the ensemble of aerosols) evaluated at the backscattering angle $\Theta = 180°$. (The Mueller matrix relates the four Stokes-vector components of the scattered beam to those of the

incident beam.) It is part of the retrieval process to obtain $\delta_l^p$ of the aerosols from the measured depolarisation $\delta_l$, where the latter is also impacted by attenuation and backscattering by molecules.

    The depolarisation ratio $\delta_l^p$ varies between 0 and 1. If $\delta_l^p = 0$, then the linearly polarized incident beam will retain its polarisation state when scattered by an angle $\Theta = 180°$. If $\delta_l^p = 1$, then the backscattered beam will be linearly polarized in the plane perpendicular to that of the incident polarization state. For spherically symmetric particles, $\delta_l^p = 0$; for nonspherical

particles, $\delta_l > 0$. Thus this quantity is a sensitive indicator for the presence of nonspherical particles. For our purposes, this allows us to discriminate between dry, nonspherical sea-salt crystals and wet marine aerosol particles in which the salt core has become dissolved in a spherical coating of water. Modelling studies suggest, indeed, that the depolarization ratio of water-covered marine aerosols goes to zero as more and more liquid water is added to the aerosol particles (Kanngießer and Kahnert, 2021b).

## 2.2 Data

  A combination of satellite based observations and reanalysis data for the June, July and August months, i.e. during southern hemispheric winter, and the 10-year period (2007-2016) is used for the evaluation. The study area is the latitude band between 40S and 65S. The Cloud-Aerosol Lidar and Infrared Pathfinder Satellite Observations (CALIPSO) Level 2, 5 km Standard Aerosol Profile product (Version 4.20) is used to obtain information on the vertical profiles of aerosol optical properties (Winker

et al., 2009, 2013). The analysis is purely based on these satellite observations and no modelling is performed. The original Algorithm Theoretical Basis Documents on how these retrievals were performed can be found in the links given below.

    https://www-calipso.larc.nasa.gov/resources/pdfs/PC-SCI-202.Part1_v2-Overview.pdf

    https://www-calipso.larc.nasa.gov/resources/pdfs/PC-SCI-202_Part2_rev1x01.pdf

    https://www-calipso.larc.nasa.gov/resources/pdfs/PC-SCI-202_Part4_v1.0.pdf

We have used aerosol retrievals of backscatter, extinction, and depolarization provided from the most recent Version 4 data products. A detailed overview of these data products is given in Young et al. (2018) and in the following link:

    https://amt.copernicus.org/articles/special_issue903.html

    The quality control and selection of CALIOP data are done as mentioned in the table below.

| Parameter/Variable | Selection criteria/condition |
| --- | --- |
| Cloud-aerosol discrimination (CAD) score | CAD>= -100 and CAD<=-20 |
| Extinction quality flag | 0 (successful retrieval and no change in the initial lidar ratio) |
| Tropospheric AOD | <= 3.0 |
| Land/Ocean | Only open ocean, sea-ice free profiles and at least 50 km away from the ice-edge |
| Height | > 500 hPa, lower tropospheric aerosols |
| Ascending/Descending orbital tracks | Only descending, nighttime |

The rationale behind these criteria is as follows. The CAD score between (and equal to) -100 and -20 makes sure that the aerosols are identified with relatively high confidence. The extinction quality flag of 0 ensures that the retrieval converges successfully and there is no need to change the initial lidar ratio. We also investigated additional cases with extinction quality flags of 1 and 2, wherein the lidar ratio was adjusted, but their contribution was little (0.25% and 0.6% respectively) and did not make much difference to the overall statistics presented here. Since the focus here is primarily on sea-salt aerosols, we investigate profiles only over the free open ocean waters and analysed only lower tropospheric aerosols (pressure > 500 hPa), mainly to avoid upper tropospheric aerosols that may be present due to the long-range from the nearby land regions. In order to avoid potential contamination and aerosol misclassification in CALIOP-CALIPSO aerosol retrievals due to wind-blown snow/ice crystals, the data profiles over the sea ice and up to 50 km from the ice edge are not included in this analysis. We use only nighttime profiles from the descending CALIPSO tracks in order to avoid potential artifacts arising from poorer signal-to-noise ratio during daytime and because we are analysing small changes in the depolarization. The total cloud cover over the Southern Ocean ranges from 75-85% depending on the winter months and region. However not all clouds are optically thick and the CALIOP lidar can see through the thin clouds. Here, we have therefore used all-sky retrievals. In this study, we were able to retrieve aerosol optical properties for nearly 40-50% of the time.

Additionally, ERA5 reanalysis is used to get information on humidity and surface winds (Hersbach et al., 2020). We further use geopotential height information at 700 hPa from Atmospheric Infrared Sounder (AIRS) on the Aqua satellites, which flies in the same orbital track as that of CALIPSO, closely in time in the A-Train constellation of satellites. The algorithm theoretical basis document for the most recent Version 7 Level 2 and Level 3 products can be found here: https://docserver.gesdisc.eosdis.nasa.gov/public/project/AIRS/L2_ATBD.pdf. We further use the daily sea-ice concentration information from the EUMETSAT's Satellite Application Facility for Ocean and Sea-Ice (OSISAF, 2017).

Since the parameterizations of the sea salt flux to the atmosphere and of the particles' water content are primarily a function of three meteorological parameters such as surface wind speed, atmospheric relative humidity and the sea surface temperature (SST), in this study, the extinction, backscatter and depolarization ratio at 532 nm of the marine aerosols are investigated in relation to these meteorological parameters. During winter months, the Southern Annular Mode (Antarctic Oscillation) plays an important role in driving meteorological variability in the southern high latitudes. The impact of these phases and the strengths of SAM can be geographical heterogeneous. When investigating the impact of SAM on aerosol properties, compiling statistics over the entire study area (40S-65S) could therefore be either misleading or could dampen the potential signal (due to the presence of pockets of low and high pressure zones over the Southern Ocean). The signal of SAM is visible in the 700 hPa geopotential heights (GPH) (Carvalho et al., 2005; Reason and Rouault, 2005). We therefore computed the $25^{th}$ and $75^{th}$ percentile thresholds of GPH at each 1x1 degree grid and used them to test if the aerosol properties respond to the two very different regimes, i.e. when the cyclonic and anticyclonic conditions prevail locally. The 700 hPa GPH distribution corresponding to both these states and the climatological state is shown in Fig. A1 in the Appendix-A. Southern Oceans are predominantly a region of intense extratropical cyclonic activity. More than half of the winter cyclones have a structure that extends through to the lower atmospheric levels (Simmonds and Keay, 2000; Houghton et al., 2001; Lim and Simmonds, 2002, 2007). Hence, the 25th percentile of GPH that corresponds to cyclonic conditions is representative of the mean state.

We compiled and investigated the joint histograms of aerosol properties, mentioned above, when a) the GPH is less than $25^{th}$ percentile for cyclonic state and b) the GPH is greater than $75^{th}$ percentile for anticyclonic state, signalling these two opposite regimes. In other words, cyclonic and anticyclonic conditions are respectively associated with the ascending and descending air masses.

## 3 Results and discussions

### 3.1 Aerosol properties in relation to wind speed and relative humidity

The 2D joint histograms of total aerosol backscattering coefficient at 532 nm as a function of surface wind speed and relative humidity are presented in Fig. 1 for three atmospheric states - mean state (Clim) (a) and when ascending (P25) (b) and descending (P75) (c) air masses are encountered. Backscattering coefficient depends both on the number density and on the size of the particles, as well as on complex refractive index and shape of the particles.

A clear demarcation in the aerosol backscatter at RH ~60% can be seen irrespective of the state of the atmosphere above which the backscatter increases. This is mainly because of the water uptake by the wind driven sea salt aerosols at high RH near the ocean surface resulting in an increase in size. An increase in both the number concentration and the mean size of sea salt aerosols contribute to an increase in aerosol backscatter values ranging from 0.005-0.012 $km^{-1}sr^{-1}$ at wind speeds below 14 m/s to above 0.018 $km^{-1}sr^{-1}$ (reaching as high as 0.025 $km^{-1}sr^{-1}$) at wind speeds stronger than 14 m/s. For a drier atmosphere (when RH < 60%), the dependency of backscatter on wind speed is negligible. A plausible reason is that such low humidities are mainly encountered at higher altitudes, at which the wind speed does not impact the emission of marine aerosol. By contrast, higher values of RH are found near the ocean surface, where wind speed directly impacts the size and number density of emitted particles, thus the backscattering coefficient.

When comparing the ascending (P25) and descending (P75) air masses, an increase in aerosol backscatter is observed in descending air masses for high RH values. This is expected as the large-scale descent inhibits vertical mixing in the atmospheric column thereby resulting in aerosols being confined to the boundary layer and their proximity to the ocean surface facilitates the growth of the particles. This is seen more clearly in Fig. 1(d) where the aerosol backscatter is expressed in terms of wind speed averaged over our study area. The mean state given by the black curve is very similar to when cyclonic conditions prevail (magenta curve). This is because the mean state is dominated by such weather systems during the SH winter months. At low and moderate wind speeds, all three curves are similar (i.e. Clim, P25 and P75). However, at higher wind speeds, elevated aerosol backscatter is observed during anticyclonic conditions or when descending air masses are encountered. This is because under such conditions emitted aerosols often remain close to the surface, where the air tends to be more humid, so that the aerosols remain moist. During cyclonic conditions, aerosols in ascending air masses are more frequently exposed to drier air, resulting in evaporation and shrinking of the particles, which reduces the backscattering coefficient.

The aerosol extinction for the aforementioned three weather states are shown in Fig. 2. Aerosol extinction is a measure of attenuation of light by absorption and scattering. A gradual increase in the extinction coefficient with wind speed with values below 0.15 is observed for RH > 60% up to 14 m/s beyond which there is an exponential increment. This can be clearly seen

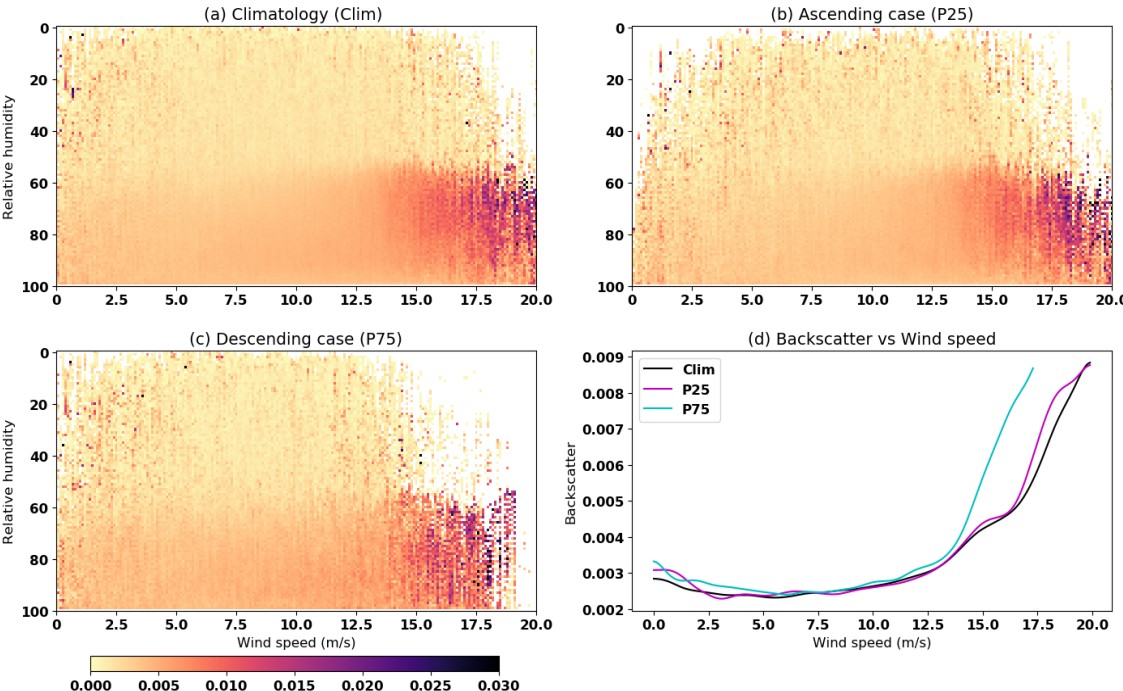

**Figure 1.** Total backscatter coefficient at 532 nm ($\mathrm{km^{-1}sr^{-1}}$) as a function of surface wind speed and relative humidity. a) Climatological conditions (Clim), all aerosol profiles included, b) Ascending case, only when 700 hPa geopotential height is less than the $25^{th}$ percentile (P25) and c) Descending case, only when 700 hPa geopotential height is greater than than $75^{th}$ percentile (P75). d) Expressed only as a function of wind speed (averaged over all relative humidities).

in Fig. 2(d) where aerosol extinction is expressed in terms of wind speed for the three cases. At stronger wind speeds (above 14 m/s), the extinction increases to values above 0.2 $\mathrm{km^{-1}}$ for higher relative humidities (RH > 60%). This is because higher wind speeds indicate more scattering sea salt particles in the atmosphere and large-sized particles which goes hand in hand
with the increased backscattering coefficients seen in the previous figure.

The observed distribution of linear depolarization at 532 nm as a function of wind speed and relative humidity is shown in Fig. 3. As seen in aerosol extinction and backscatter parameters, a clear distinction at RH ~60% is observed here too. At low RH, for RH less than 60%, the linear depolarization values are clearly higher indicating a dominance of more non-spherical sea salt particles. Remarkably enough, the linear depolarization is not very sensitive to varying winds. Modelling
studies suggest that the depolarization ratio of sea-salt particles is dependent on particle size (Kanngießer and Kahnert, 2021a). This size-dependence would translate into a pronounced wind-speed dependence, which is not evident in the observations. The high depolarization cases are more likely from the free troposphere which are decoupled from the surface. It is also evident that there are more non-spherical particles when cyclonic conditions prevail compared to the climatological mean state

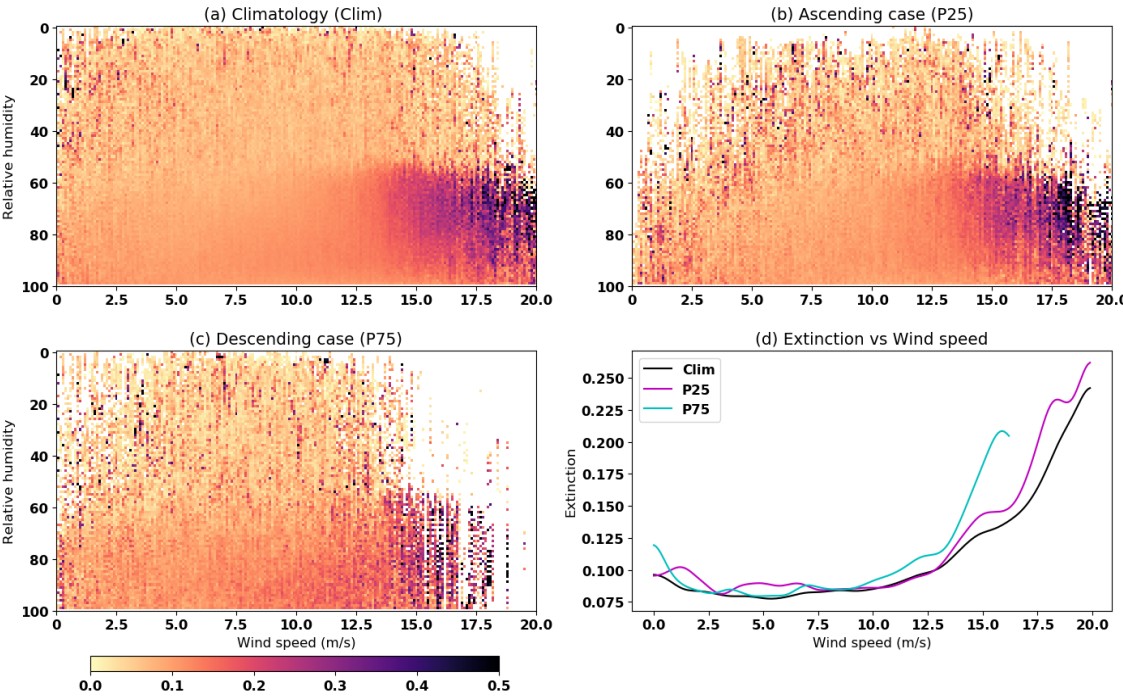

**Figure 2.** Same as in Fig. 1, but for the extinction coefficient (km$^{-1}$) at 532 nm.

and anticyclonic conditions. This is as expected since the aerosol particles tend to dry out in ascending air parcels as the RH
decreases with height. As the RH increases (RH > 60%), water uptake by hydrophilic sea salt particles and subsequent rounding
results in lower depolarization values as can be seen in all the three cases. Lab and in-situ measurements have shown that the
efflorescence (crystallization) of sea salt occurs at a range of 46-58% RH (Zeng et al., 2013) and deliquescence (the onset of
water absorption) occurs at RH around 70-75% (Tang, 1997; Zieger et al., 2017) when the particle starts losing its crystalline
structure and tends to be more spherical resulting in lower depolarization values. These RH numbers seem to agree well with
what is seen over the Southern Oceans.

The dependency of linear depolarization on wind speed when the air parcel is classified as 'wet' (solid lines) and 'dry'
(dashed lines) conditions based on relative humidity being above and below 60% respectively is shown in Fig. 3(d). This is
shown for all three weather states. It is evident that the linear depolarization ratio is significantly higher in dry air masses, indi-
cating the presence of more non-spherical particles compared to moist environments where the particles have a more spherical
shape due to water uptake. Also, lower depolarization ratios are observed in ascending air masses compared to descending air
masses due to the fact that RH decreases as we move away from the surface. Over the Southern Oceans, inversions in RH were
not detected. A slight increase in depolarization ratios with wind speed is noticeable in moist environments.

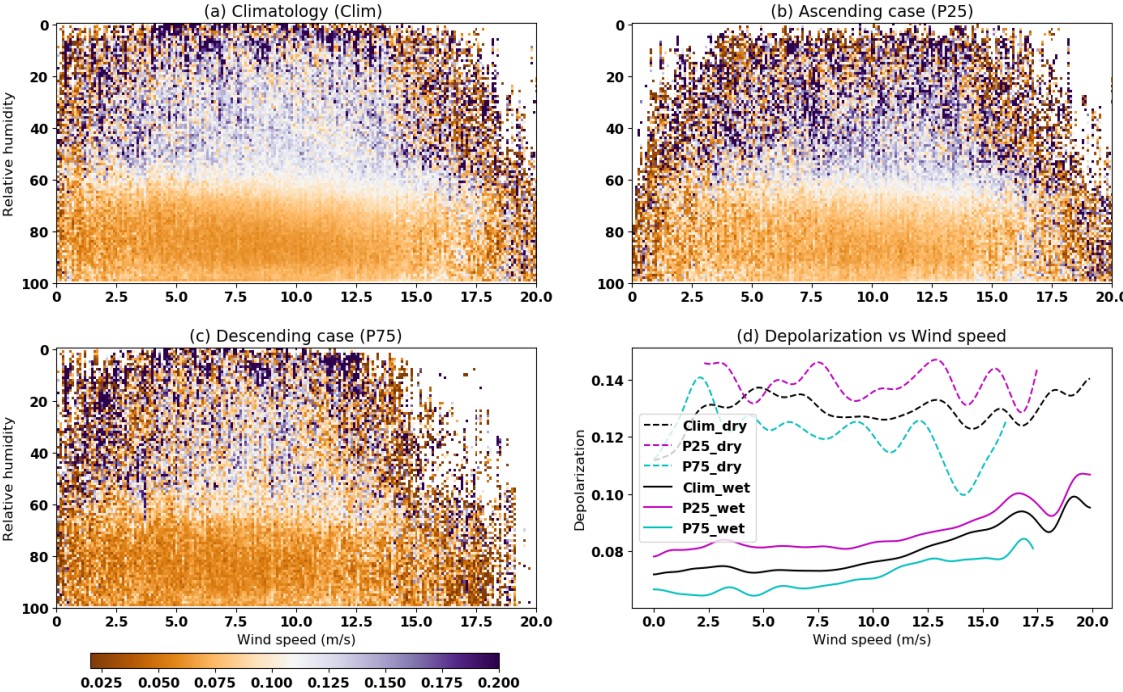

**Figure 3.** Same as in Fig. 1, but for the particulate depolarization ratio (532 nm). In d) depolarization is further classified in two categories based on the relative humidity, namely dry (RH < 60%) shown as dashed lines and wet (RH > 60%), shown as solid lines and expressed as a function of surface wind speed.

## 3.2 Aerosol properties in relation to sea surface temperatures

Marine aerosol concentration in the atmosphere may also depend indirectly on the temperature by facilitating the evaporation
of the sea salt bubbles, if the bubbles are small in size or by accelerating the bursting process with an increase in temperature. Studies have shown that the nano-sized particle concentration decreases with increasing sea water temperature and for particle sizes greater than 0.1 $\mu$m, the particle concentration increases with increasing sea water temperature (Nilsson et al., 2007; Martensson et al., 2003). Here, we analyze the optical properties of the marine aerosols over the Southern oceans in relation to wind speed and sea surface temperature. The 532 nm aerosol backscatter and linear depolarization ratio are shown in Fig.
4 and Fig. 5 respectively for the three weather states. In this case, we investigated only the three lowermost quality controlled aerosol profile bins nearest to the surface as they are expected to be influenced most by the SST driven emissions. In the upper layers of the atmosphere, the SST is unlikely to have a direct influence on the backscatter and extinction.

Increased aerosol backscatter is observed at higher wind speeds in the climatological mean state and when cyclonic conditions are encountered. We see in Fig. 5 that under cyclonic (ascending, P25) conditions the aerosols depolarise more strongly
than under anticyclonic (descending, P75) conditions. This indicates that even within the lowest three altitude bins considered

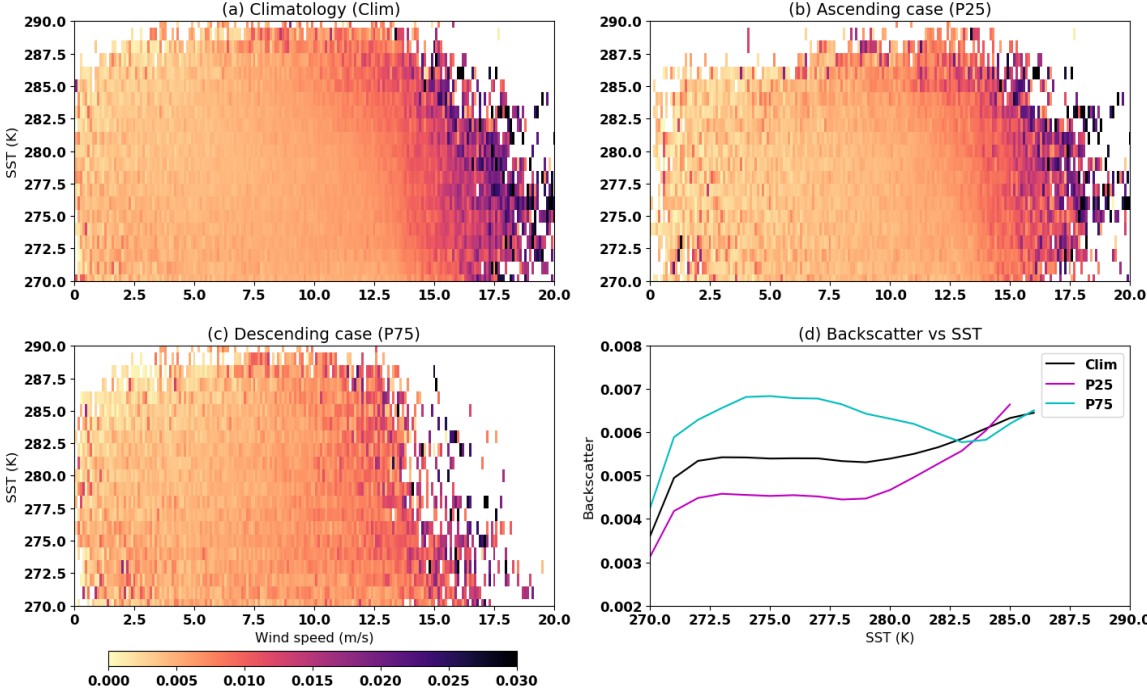

**Figure 4.** Total backscatter coefficient at 532 nm ($\mathrm{km}^{-1}\mathrm{sr}^{-1}$) as a function of surface wind speed and temperature. a) Climatological conditions, all aerosol profiles included, b) Ascending case, only when 700 hPa geopotential height is less than $25^{th}$ percentile (P25), c) Descending case, only when 700 hPa geopotential height is greater than than $75^{th}$ percentile (P75). d) Expressed only as a function of temperature (averaged over all wind speeds.

in this figure the aerosols can partially dry out in ascending air parcels, resulting in higher values of the depolarization ratio. Accordingly, the loss of water reduces the particles' geometrical cross section, resulting in a decrease in their backscattering cross section as compared to descending aerosols (see Fig. 4). However, a dependency on the SSTs is very weak for lower wind speeds in all three weather states. At wind speeds exceeding about 12 m/s, we observe that the backscattering coefficient generally increases with SST. This is consistent with increasing emissions of marine aerosols with increasing SST, as reported in Nilsson et al. (2007); Martensson et al. (2003). When averaged over all wind speeds, the initial 2.5 deg increment (from 270 K to 273 K) in the SST increases the aerosol backscatter coefficient by 30 - 40%, but remains a constant for warm SSTs (i.e. above 273 K) up until about 280 K, beyond which we observe, again, an increase in the backscattering coefficient. There may be several competing effects responsible for this rather complex behaviour. Higher SST does not only increase the number density of emitted particles, but it also favours the emission of particles with larger diameters (Martensson et al., 2003). Both effects contribute to an increase in the backscattering coefficient. However, larger particles are also more likely to be re-deposited to the ocean surface, thus reducing the backscattering signal. This effect can be expected to be more efficient under descending

conditions (P75), under which the particles are more confined to the surface as well as heavier, because they are less likely to dry out. Indeed, we do observe a decrease in the backscattering coefficient for descending (P75) conditions between 275–284 K. In the climatological case and under ascending conditions (P25), this effect only dampens the increase of the backscattering coefficient with SST, but it is not pronounced enough to result in a net decrease. Similar features are also seen in the linear depolarization ratio (Fig. 5). The values increase when temperatures rise from 270 to 273 K and thereafter plateaus until 280 K. Under descending conditions (P75) the particles remain moist and mostly spherical. Accordingly, the depolarisation is very low, around 0.045, and does not change with SST>273 K. Under ascending conditions, the particles partially dry out, resulting in depolarisation ratios up to 0.075. (This is much lower than the maximum values around 0.15 that we observed in Fig. 3; but recall that we are only considering the lowest three altitude bins here.) At SST>278 K, the depolarization ratio drops under ascending conditions (P25). One possible hypothesis to explain this effect is that the size distribution of the emitted marine aerosol particles shifts to larger particle diameters with rising SST (Martensson et al., 2003). But larger particles need more time to dry out. Thus, one would expect that, at least near the surface, higher values of SST can result in less depolarisation under ascending conditions, because evaporation of water in ascending air masses and production of nonspherical, depolarizing particles will be slower for large particles than for small particles.

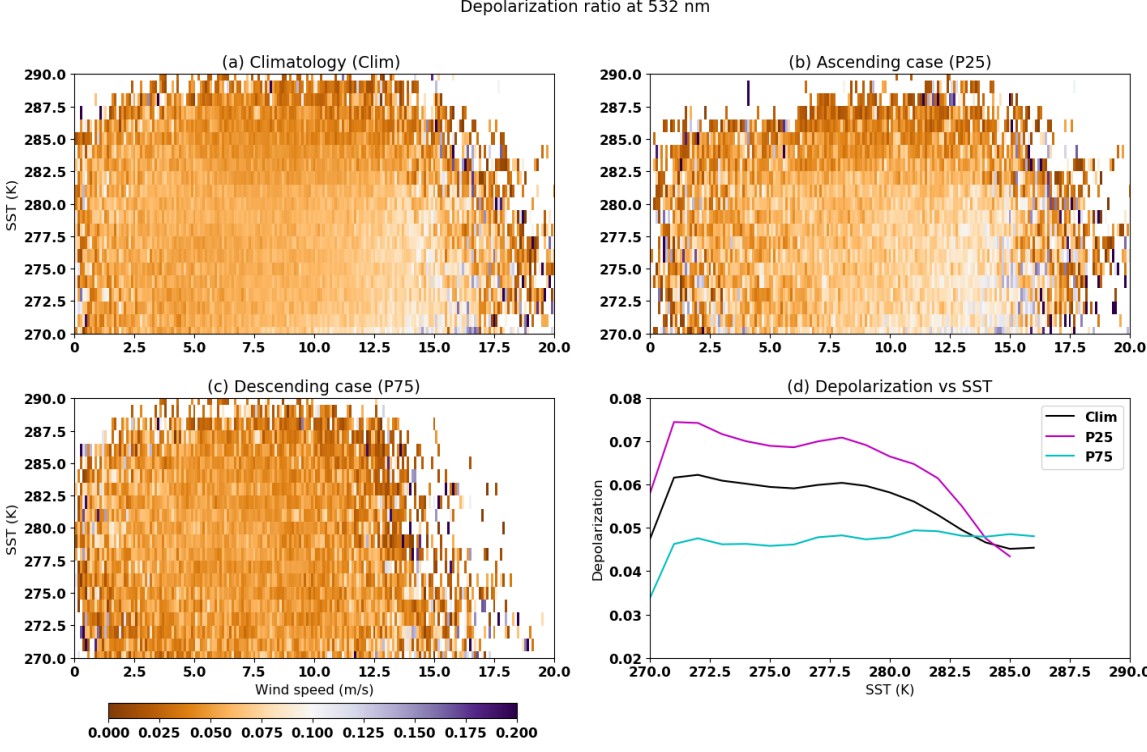

**Figure 5.** Same as Fig.4, but, for the total linear depolarization ratio (532 nm) as a function of surface wind speed and temperature.

### 3.3 Verticality of aerosol properties in relation to relative humidity

The vertical distribution of backscatter coefficient as a function of RH is presented in Fig.6 for climatological mean state and in ascending (P25) and descending (P75) air masses. In the case of P25 conditions, there are not many observations where RH is lower than 40% in the lowermost troposphere as expected. Aerosol backscatter values in the free troposphere are much lower than in the boundary layer, irrespective of the RH and the three weather states. This indicates that local emissions from the ocean surface make the dominant contribution to aerosol loads over the Southern Ocean. We also see a pronounced dependence of backscattering on RH up to 850 hPa. Elevated values of RH entail a higher aerosol water content, hence larger mean aerosol radius, which results in higher values of the backscattering coefficient. This effect is very pronounced up to 850 hPa. In the free troposphere, no clear dependence of backscattering on RH can be seen. It is possible that the long-range transported aerosols above 850 hPa consists of a mixture of different aerosol species with varying water-adsorption properties. But most importantly, the aerosol in the free troposphere is too tenuous, and the detected signal is too weak to discern a clear relation between RH and backscattering. It is also notable that for any fixed value of RH > 50%, the aerosol layer near the surface has a higher vertical extent during cyclonic conditions (P25) than during anticyclonic conditions (P75), which is what we already mentioned in the discussion of Fig.1. During cyclonic conditions, the lower atmospheric layers are significantly wetter which increases the backscattering coefficients for reasons mentioned above.

The behaviour of backscattering coefficients with RH in the free troposphere (above 850 hPa) and in the lower troposphere are shown in Fig.6d. While there is only a gradual monotonous increase in the backscattering values relative to RH in the free troposphere, a significant increase is observed in the lower troposphere for relative humidities above 40 %. This is, again, an indication that the free troposphere may contain aerosols other than sea salt, which respond differently to changes in RH, while the lower troposphere is likely to be completely dominated by marine aerosol. Note that in the P25 case, there are very few observations available below 850 hPa where RHs are lower than 60% leading to an artifact where there is a sudden increase in backscatter at the lower end. We focus now on the lower troposphere (solid lines) and values of RH > 50%. At any given value of RH, we observe higher backscattering values for cyclonic (P25) than for anticyclonic conditions (P75). One possible explanation would be that elevated wind speeds with high aerosol emissions are more prevalent during cyclonic conditions.

The variability of aerosol extinction in the vertical in relation to RH evaluated as joint histograms is shown in Fig.7. All the three weather states show similar features with varying magnitude. We clearly see two maxima in aerosol extinction - in the lower troposphere and in the upper troposphere and the intensity of attenuation increases with increasing RH, more steeply in the lower troposphere. This indicates that the aerosol layer in the free troposphere has different hygroscopicity properties, which can be interpreted as another indication for the presence of long-range transported aerosols aloft that also contain aerosol species other than sea salt. This is further corroborated by the fact that the presence of aerosols in the free troposphere is evident in the extinction profiles, but not in the backscattering profiles. This indicates that the aerosols in the free troposphere may be more strongly absorbing than typical marine aerosols. The increase in extinction with RH seen in the lower troposphere is similar to what is observed in aerosol backscatter. The sharp increase in attenuation occurs at 50 % RH and is associated with

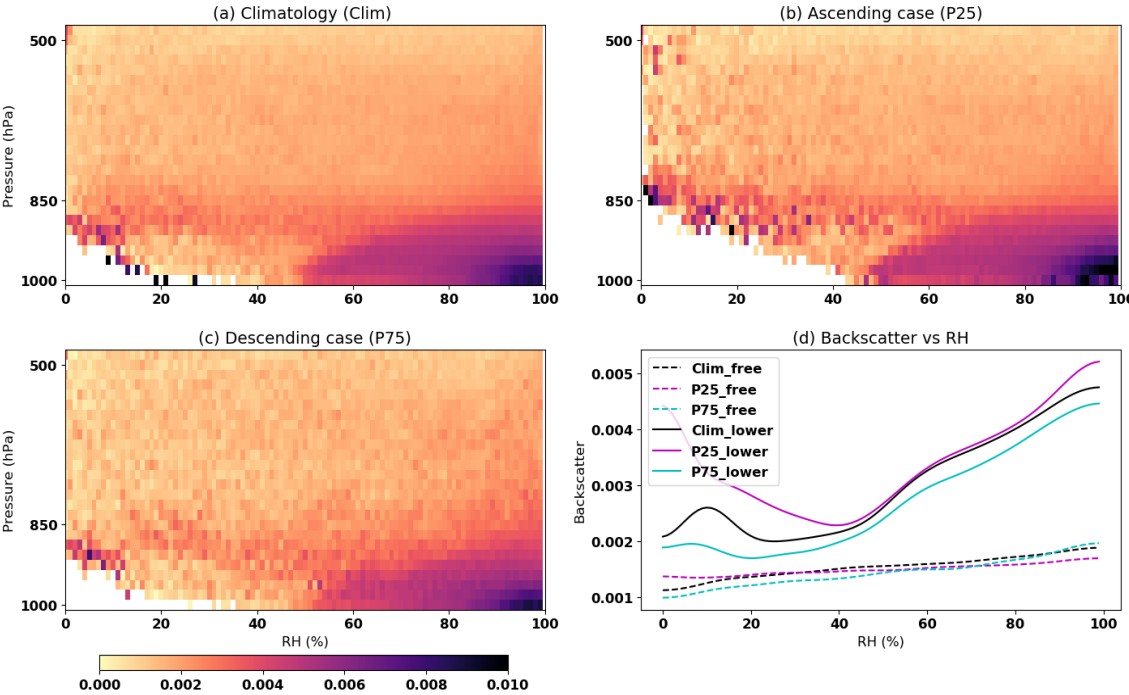

**Figure 6.** The vertical structure of the total backscatter coefficient at 532 nm ($km^{-1}sr^{-1}$) as a function of relative humidity. a) Climatological conditions, all aerosol profiles included, b) Ascending case, only when 700 hPa geopotential height is less than $25^{th}$ percentile (P25), c) Descending case, only when 700 hPa geopotential height is greater than than $75^{th}$ percentile (P75), d) Expressed only as a function of relative humidity for two cases; the free tropospheric aerosols (pressure < 850 hPa) and the lower tropospheric aerosols (pressure > 850 hPa).

the presence of large aerosol particles. As expected, the extinction coefficient values are slightly less in descending air masses compared to the cyclonic cases.

     The extinction curves for varying relative humidities when partitioned into lower and free troposphere cases for the three weather states is shown in Fig.7d. In the lower troposphere a steady increase in extinction and a sharp increase beyond 50% RH is observed in all the cases. Extinction by marine aerosols is dominated by scattering. Thus, the extinction curves in the

lower troposphere display a dependence on RH that is very similar to that of the backscattering curves in Fig. 6d. In the upper troposphere, the dependence of extinction on RH is much weaker, which we attribute to the presence of other, presumably less hygroscopic aerosol species. By contrast to the lower troposphere, the aerosol extinction is higher in the P75 case compared to the other two states in the free troposphere.

     While backscatter coefficient and extinction coefficient are indicative of the presence of aerosols and their size, linear de-

polarization ratio gives information about the morphology of the aerosols, i.e. if the particles are spherical or nonspherical. Dry sea salt particles have shapes similar to cubes, which can have depolarisation ratios up to, but rarely exceeding 0.20 (see

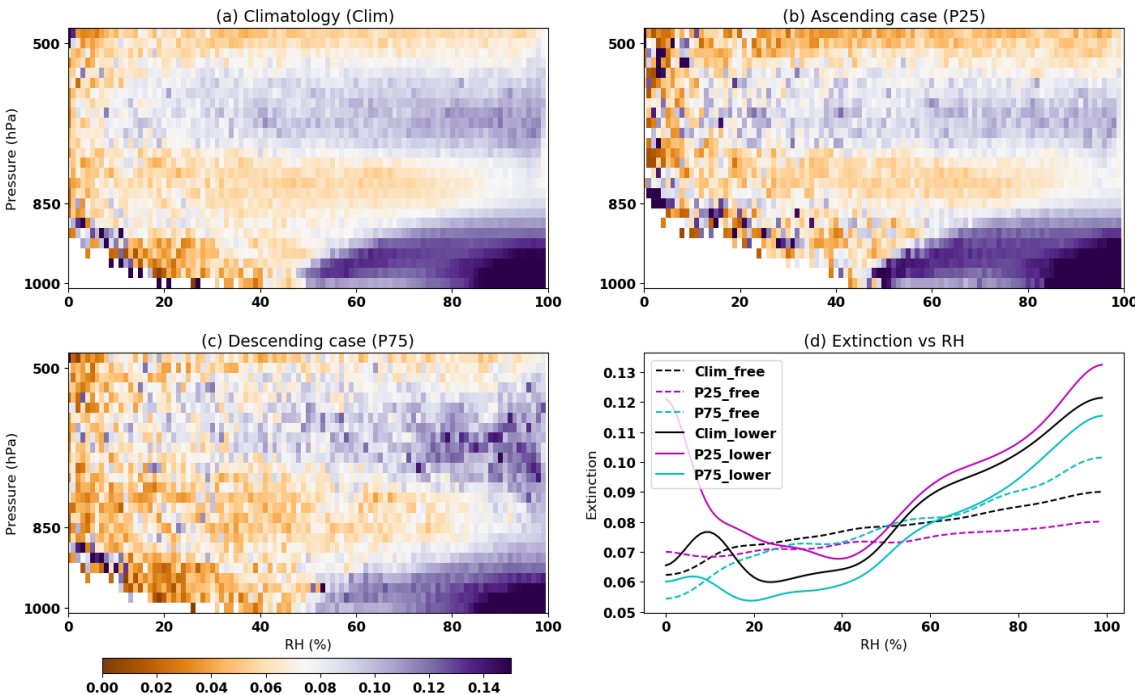

**Figure 7.** Same as in Fig. 6, but for the extinction coefficient (532 nm).

Kanngießer and Kahnert (2021a), and references therein). As more water is adsorbed, the particles become more spherical, and the salt core is partially or completely dissolved in the liquid coating. The depolarisation ratio of homogeneous spheres is identically zero. Thus the depolarisation ratio should anti-correlate to the water content of the particles, which, in turn, depends on RH.

As seen in the earlier section, RH is one of the major factors that can impact the depolarization ratio of a particle. Here, we analyze the vertical distribution of this ratio as a function of RH for the three weather states (Fig.8). A clear separation is observed in the free and lower troposphere, more prominent in the climatological mean and the cyclonic states. For RH > 60%, low depolarisation values are noticeable in the lower troposphere, which is an indication of the dominance of water-coated, 370   mostly spherical sea salt particles. For RH < 60%, there are instances when the aerosol depolarization increases in the boundary layer, more prominently in the mean and anticyclonic cases which can be associated with the presence of drier aerosol particles.

   As seen by comparison with Fig. 7, the region of high values of aerosol extinction in the upper troposphere coincides with high depolarization values irrespective of the RH values. Again, this indicates the presence of long-range transported aerosols of mixed composition, which evidently have water adsorption properties different from those of pure marine aerosols. However, 375   the composition of these aerosols is unknown, and it may, indeed, be highly variable in space and time. It is, therefore, much more difficult to interpret this data. Fig. 8d shows the depolarisation ratio as a function of RH, averaged over the geographic

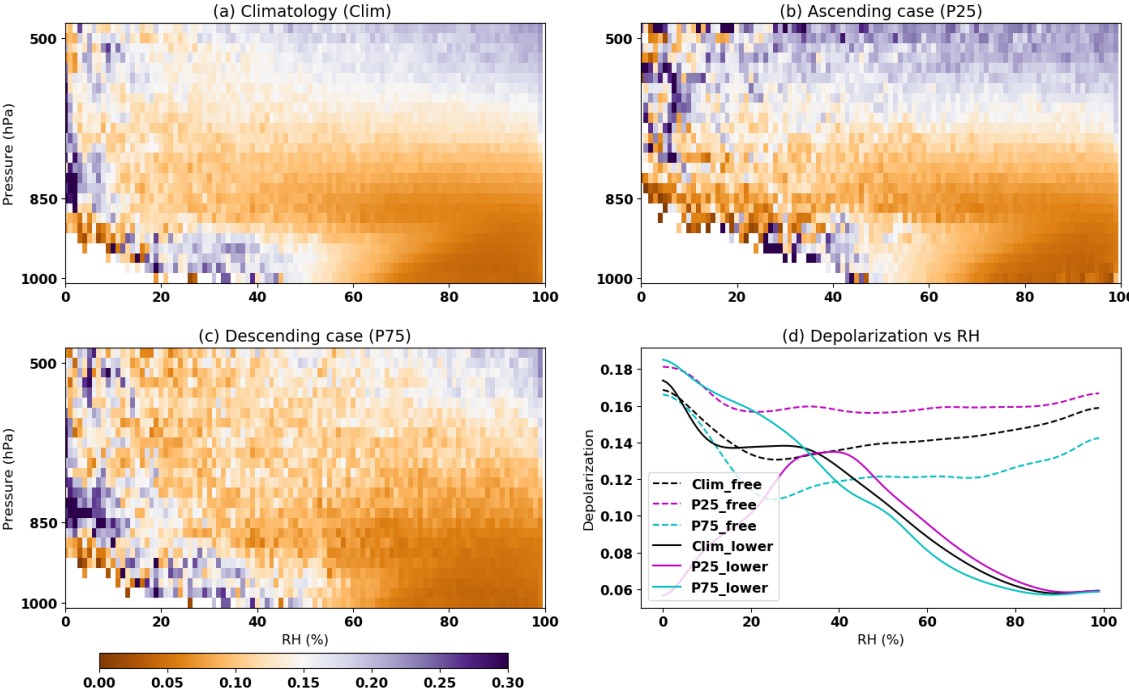

**Figure 8.** Same as in Fig.6, but for the particulate depolarization ratio (532 nm).

domain and the study period. . High depolarization values at the lower end of the RH spectrum indicate the presence of non-spherical particles. For 40% < RH < 100% the depolarisation in the lower troposphere is characterised by a steep but gradual decrease with increasing RH. This is true for all three weather states. However, we do not see any sudden changes in the

depolarisation ratio. One may have expected such changes at the efflorescence point or at the deliquescence point, at which pure sodium chloride could undergo rapid changes in water content, hence in shape. We can conclude that the decrease of the depolarisation ratio with increasing RH in the lower troposphere is consistent with a change in sea salt morphology from nonspherical to spherical shape, which is caused by increased water adsorption with growing RH. However, our averaged data set does not display any evidence of sudden crystallisation, deliquescence, or hysteresis effects in water content. This does not

rule out the existence of such effects; but discovering them will likely require a targeted investigation of individual cases, rather than an analysis of temporally and geographically averaged large data sets. We note that for RH < 35% the P25-curve diverges from the climatological and the P75-curve. However, such low RH-values during cyclonic conditions are rather rare. Thus, this part of the curve is based on fairly sparse data and may well be an artifact.

In Fig.9, the vertical distribution of depolarization ratios is analyzed for two ranges of relative humidities that characterizes

drier and wetter atmospheric conditions over the Southern Oceans. The RH range between 20 and 60% is considered as 'dry' and when RH exceeds 60%, those air parcels are considered 'wet'. The three curves show the three weather states. It can be

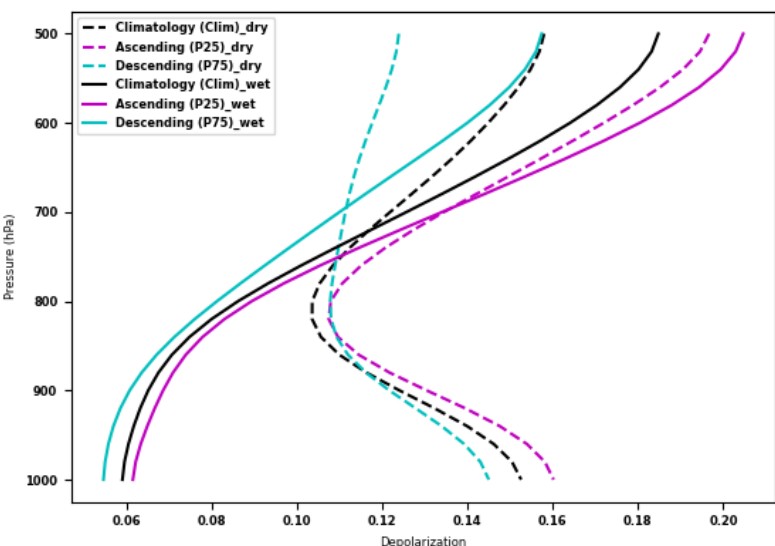

**Figure 9.** The vertical structure of depolarization ratio for two relative humidity groups; when relative humidities are between 20% to 60% (drier conditions, dashed lines) and when they are between 60% - 100% (wetter conditions, solid lines). The three colours show climatological, ascending (P25) and descending (P75) conditions.

clearly seen that the drier air masses always have substantially higher depolarization ratios in the lower atmospheric levels than the moist aerosols. However, this ratio decreases with height until 800 hPa and then increases. This behaviour can also be seen in the colour panels in Fig. 8. As mentioned earlier, we have two distinct aerosol layers, one marine aerosol layer near the surface and one mixed layer in the free troposphere. The region around 800 hPa marks the boundary between these layers that is characterised by low aerosol concentrations, which explains the minimum in the depolarisation curve under dry conditions. By contrast, owing to the predominance of spherical shape, the wet aerosols have a low depolarization ratio, despite their high concentration near the surface. Irrespective of the weather state the ratio increases with height indicating more non-spherical aerosols due to lack of moisture resulting from a decrease of RH in the vertical. The wet aerosols encountered during cyclonic conditions have higher depolarization ratios compared to those encountered during anticyclonic conditions. Again, above 800 hPa, where we expect to find long-range transported mixed aerosol populations, the interpretation of the depolarisation ratio and its dependence on the vertical coordinate and RH is rather elusive.

### 3.4 Fitting functions for the aerosol optical properties based on the observational analysis

Based on the analysis presented here, we propose simple statistical parameterizations for the aerosol optical properties as a function of the meteorological variables discussed here for climatological conditions. These statistical relationships can be used to evaluate the corresponding sea salt parameterization schemes used in climate models. For the aerosols below 850 hPa,

optimal polynomials are fitted to express the dependency of aerosol backscatter and extinction on wind speed, relative humidity and temperature using the least squares approach. Equations (3) and (4) define respectively aerosol extinction and backscatter as a function of wind speed and relative humidity, whereas equation (5) defines aerosol backscatter as a function of wind speed and sea surface temperature. These parameterizations are:

$$ext_{ws}^{rh} = a3 * ws^3 + a2 * ws^2 + a1 * ws + b2 * rh^2 + b1 * rh + c \tag{3}$$

$$bak_{ws}^{rh} = a3 * ws^3 + a2 * ws^2 + a1 * ws + b2 * rh^2 + b1 * rh + c \tag{4}$$

$$bak_{ws}^{T} = a3 * ws^3 + a2 * ws^2 + a1 * ws + b2 * T^2 + b1 * T + c \tag{5}$$

where, ext, bak, ws, rh and T are aerosol extinction and backscatter (532 nm), wind speed (m/s), relative humidity (%) and sea surface temperature (K) respectively. The values of coefficients are provided in Table 1 below. Note that these dependencies are prepared for boundary layer aerosols (below 850 hPa), for RH>50% and for ocean and ice-free surfaces.

| Coefficients | $ext_{ws}^{rh}$ | $bak_{ws}^{rh}$ | $bak_{ws}^{T}$ |
|---|---|---|---|
| a3 | -7.26389640e-06 | -1.83516997e-08 | 3.34003672e-06 |
| a2 | 8.91368900e-04 | 2.65490114e-05 | -2.05591699e-05 |
| a1 | -6.77922228e-03 | -2.02947001e-04 | 7.76695055e-05 |
| b2 | -9.48527177e-05 | -4.66224393e-06 | -4.12577937e-06 |
| b1 | 1.47082752e-02 | 7.31508504e-04 | 2.39795430e-03 |
| c | -4.54235227e-01 | -2.43601733e-02 | -3.43857316e-01 |

**Table 1.** Coefficients values required for parameterizations shown in Eq 3-5.

We further parameterized the dependency of aerosol depolarization on all the observed ranges of relative humidity. The depolarization decreases monotonically as the relative humidity increases. However, this decrease occurs in a stepwise function (Fig. 8). Therefore, the following parameterization is proposed to represent this dependency.

$$\delta^{rh} = a * exp^{-(b3*rh^3 + b2*rh^2 + b1*rh)} + c \tag{6}$$

where, a = 1.15838914e-1, c = 5.91111849e-02 and b3, b2 and b1 are 1.96280779e-05, -1.48341788e-03 and 4.13361512e-02 respectively.

## 4   Conclusions

70% of the Earth's area is covered by water. The marine aerosols, especially sea salt, are therefore one of the dominant sources of the total aerosol load in the atmosphere. The significant climate implications of marine aerosols call for better understanding of their variability and coupling with the local meteorological conditions. The Southern Ocean is unique in this context, as the boundary layer is relatively free from the influence of anthropogenic aerosols and dust (compared to the northern hemispheric oceans). During the southern hemispheric wintertime the influence of natural DMS emissions on the total aerosol load is also negligible over the Southern Ocean, allowing investigations of sea salt aerosols. Using 10 years of wintertime (June, July and August, 2007-2016) space based profiling of aerosols by CALIOP-CALIPSO in combination with meteorological reanalysis data, we investigated the sensitivity of marine aerosol properties over the Southern Ocean to various meteorological parameters, such as relative humidity, near surface wind speed, and sea surface temperature in terms of joint histograms. The sensitivity study is done for the climatological conditions as well as during the enhanced cyclonic (P25) and anticyclonic conditions (P75) in order to understand the impact of large-scale atmospheric circulation on the aerosol properties. These conditions were defined based on the 700 hPa geopotential height being either below $25^{th}$ percentile (P25, cyclonic, ascending air masses) or above $75^{th}$ percentile (P75, anticyclonic, descending air masses).

The following conclusions are drawn based on this sensitivity analysis.

a) A clear demarcation in the 532 nm aerosol backscatter and extinction at RH ~60% can be seen irrespective of the state of the atmosphere. The backscatter and extinction increase at higher relative humidity as a function of wind speed. This is mainly because of the water uptake by the wind driven sea salt aerosols at high RH near the ocean surface resulting in an increase in size, which is confirmed by the decreased depolarization for the wet aerosols.

b) An increase in aerosol backscatter and extinction is observed during the anticyclonic conditions compared to cyclonic conditions for the higher wind speeds and relative humidity, mainly due to aerosols being confined to the boundary layer and their proximity to the ocean surface facilitates the growth of the particles.

c) At lower wind speeds, there is a very weak dependency of aerosol backscatter on the SSTs. At wind speeds exceeding about 12 m/s, the backscattering coefficient generally increases with SST.

d) When aerosol properties are investigated in terms of aerosol verticality and in relation to meteorological parameters, it is seen that the aerosol backscatter values in the free troposphere (pressure < 850 hPa) are much lower than in the boundary layer, irrespective of the RH and the three weather states. This indicates that the local emissions from the ocean surface make the dominant contribution to aerosol loads over the Southern Ocean.

e) We see a pronounced dependence of backscattering on RH up to 850 hPa. Elevated values of RH entail a higher aerosol water content, hence larger mean aerosol radius, which results in higher values of the backscattering coefficient. This effect is very pronounced up to 850 hPa. In the free troposphere, no clear dependency of backscattering on RH can be seen.

f) A clear separation of linear depolarization is observed in the free and lower troposphere, more prominent in the climatological mean and the cyclonic states. For RH > 60%, low depolarization values are noticeable in the lower troposphere, which is an indication of the dominance of water-coated, mostly spherical sea salt particles. For RH < 60%, there are instances when the aerosol depolarization increases in the boundary layer, more prominently in the mean and anticyclonic cases which can be associated with the presence of drier aerosol particles. Under very dry conditions, the depolarisation ratio of marine aerosols in the boundary layer can be as high as 0.16.

g) Interpreting the verticality of aerosol properties (especially depolarization) in relation to the meteorological conditions in the free and upper troposphere (pressure < 850 hPa) is however difficult and challenging in a statistical sense. The tracer transport studies are required to assess the contribution and mixing of anthropogenic and natural biomass burning pollutants in the free troposphere.

This study elaborates on the relative importance of RH on the optical properties of sea salt over the Southern oceans during SH winter months when other natural emissions and long range transport of anthropogenic emissions are negligible. Although a handful of in situ measurements have been carried out using ship cruises and aircrafts, this is the first study of its kind that has attempted a detailed mapping of the optical properties of sea salt in its natural environment on the basis of the three fundamental parameters that the emissions depend upon - RH, wind speed and sea surface temperature and their vertical distribution purely from satellites. The generation of sea salt aerosols in the models are described in terms of source functions that depend on either one or more of the factors mentioned above for varying size ranges and these are derived from lab or ship/aircraft measurements. These source functions are adapted depending on the model to get the emissions in the correct order so as to accurately estimate the total aerosol optical depth (for example, some models introduce a temperature dependency or alter the power factor so as to increase/decrease the wind speed dependency or include the super micron size particles in the calculations). Also, due to the high hygroscopicity of sea salt, the parameterization schemes simulate the emissions at 80% RH (Monahan et al., 1986). To introduce the variability of the water uptake by dry sea salt particles, recent models use hygroscopic growth factors at varying RH that are prescribed (Chin et al., 2002) or that are based on empirical equations or are explicitly calculated (Ghan et al., 2001; Vignati et al., 2004). The dependency of the aerosol optical properties on RH is of profound importance for climate forcing estimates, particularly, in the case of sea salt as they can co-exist in both phases at the same time between 50% and 70% RH. And this dependency changes when sea salt particles are contaminated by other chemical species. In this context, the present study quantifies the variability in the sea salt aerosol optical properties on a range of observed relative humidities, surface temperatures and wind speeds over the Southern Oceans. We further provide the fitting functions for aerosol extinction and backscatter and depolarization ratio based on one or more of the meteorological parameters such as wind speed, relative

humidity and temperature for naturally occurring sea salt which can in turn, help improve the source functions used in the sea salt parameterization schemes. Further insights are needed to map the behaviour of the optical properties of sea salt when they are coated with other naturally occurring chemical components as is in the case of the rest of the oceans.

*Data availability.*

All data sets used here are publicly available. The links to the data sets are given below:

The CALIPSO Level 2 standard aerosol profile product (CAL_LID_L2_05kmAPro-Standard-V4-20) version 4.2 available at 5 km horizontal resolution is used. The data are accessible via

https://doi.org/10.5067/CALIOP/CALIPSO/LID_L2_05KMAPRO-STANDARD-V4-20 (NASA Earth Data, 2020); created by NASA Langley Atmospheric Science Data Center DAAC.

The AIRS satellite version 7 dataset is used for the 700 hPa geopotential height and can be accessed via https://disc.gsfc.nasa.gov/datasets/. The data were processed at the Jet Propulsion Laboratory, California Institute of Technology (https://airs.jpl.nasa.gov/data/products/v7-L2-L3/).

The humidity and surface winds are taken from ERA5 reanalysis (https://doi.org/10.1002/qj.3803; Hersbach et al., 2020) and are available at the Climate Data Store (CDS) via the link https://cds.climate.copernicus.eu/ (Copernicus Climate Change Service, 2017).

The sea ice concentration from EUMETSAT's OSISAF 2017 is used and is available at https://osi-saf.eumetsat.int/products/osi-401-b.

*Author contributions.*

MT carried out the analysis and wrote the first draft of the manuscript. All authors contributed to the design of experiments, interpretation of the results and writing.

*Competing interests.* The authors declare no competing interests.

*Acknowledgements.* The authors acknowledge the funding from Swedish National Space Agency (Project nr. 126/19) for this study.

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

670

## Appendix A

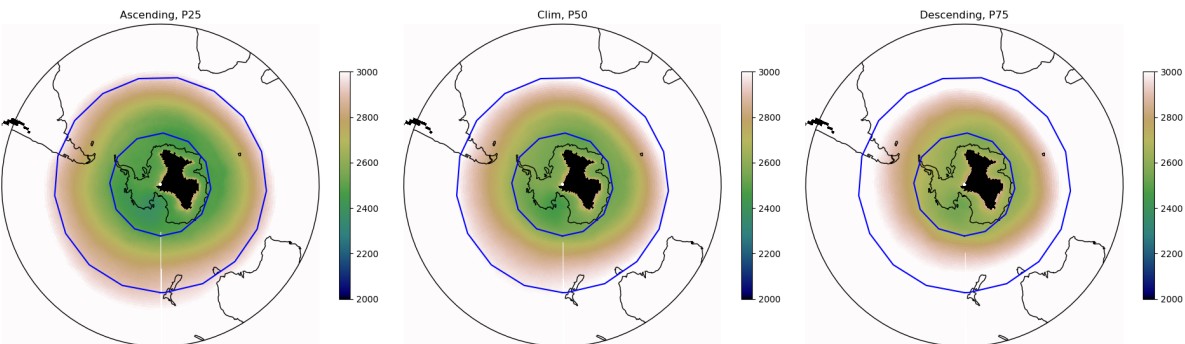

**Figure A1.** The 700 hPa geopotential height distribution corresponding to the three cases (L to R): ascending (P25), climatological and descending (P75) conditions.