# Peer review of "Marine aerosol properties over the Southern Ocean in relation to the wintertime meteorological conditions"

_Atmospheric Chemistry and Physics, 2021_

## Referee Comment (RC2)

Review of "*Marine aerosol properties over the Southern Ocean in relation to the wintertime meteorological conditions*"
by Thomas et al.

I have now read the manuscript of Thomas et al. which concerns (optical) properties of the Southern Ocean marine aerosols.

My overall recommendation is major revision. Indeed, the present manuscript suffers some flaws that need to be addressed. I have also some requirements and points the authors must clarify. Below are some comments.

- The authors should present a geographic map to locate the investigated area(s).
- An important reference (to be quoted) about marine aerosols in the Indian Southern Ocean is Mallet et al. (2018) "*Marine aerosol distribution and variability over the pristine Southern Indian Ocean*" . Atm. Env. 182.

l. 41 : interference → influence
l. 104 : Indicate what is $u\_10^{3.5}$. It is clear but explicit indication would help some readers
l. 143 "their contribution was little and did not make much difference to" : This must be quantified
l. 146: "The profiles over the sea-ice and within 50 km from the ice-edge are removed to avoid potential contamination and misclassification due to the wind-blown snow/ice crystals." : This sentence is unclear and must be rephrased.
l. 167: How did the authors obtain the backscatter coefficient? And, later, the extinction coefficient? Data? Model? Is it a coefficient or a cross section? Particles are supposed to be spherical in the calculation?
P25 and P75 must be explicitely specified. 25%ile and 75%ile are not standard notations.
l.170: Is it the optical index or the refractive index (real part of the optical index)?
l. 175 and around: The considerations given must be quantified.
Fig.1 and others: It is quite strange to have a decreasing axis of RH. I assume it is because absolute humidity is higher close to the sea. It is not necessary the same for RH since it depends on temperature.
l.190-195: The authors talk about the aerosol extinction, so do they really include absorption in their discussion? Sea salt are not very absorbing. They say that 'similar conclusions can be derived for the aerosol extinction'. This must be detailed a bit more. Also, what is the relation with RH and aerosol size?
l. 212: water absorption → water uptake
Legend of Fig.4: symbol for steradian is sr and not Sr
Section 3.3: The authors use a vertical distribution. They must either include an altitude axis z (km or m) to help readers understanding or convert pressure into altitude.

It is not surprising that total backscattering increases with RH and p: RH provides water vapour and high pressure and low temperature (low thermal agitation) favor water uptake. Figure 6 is not at all unexpected. So, do the results obtained really original? This is a serious flaw of the paper: the authors present scatterplots and make only general comments on what they see on their figures. Also, it would be a clear advance in the field to propose a parameterization of the relations between RH and basckscattering and extinction (and/or absorption). Moreover, since RH contains both absolute humidity (AH) and T, the same study using AH would be useful. Such considerations are important then for modeling.
Finally, since the authors recall in their introduction that marine aerosols are important for climate change, they should estimate radiative forcing due to water uptake by marine aerosol (it is not enough for me to focus only (qualitatively) on data about RH and optical properties as done here).
l.278: cubes → polyhedrons

l.285: depolarization is interesting to discriminate dry sea salts from wet ones. Can the authors use their data of depolarization to perform such a discrimination? It is essential since both kind of aerosols do not have the same radiative (direct/indirect) effects. This would be an added value to the study presented.

In the conclusion, joint histograms are mentioned. I did not see any histogram in the manuscript.

For all of the points listed above, I do not recommend a publication before major revisions have been done.

---

## Author Response (AR1)

**Response to Reviewer #1**

This study presents a detailed investigation into sea spray aerosol properties over the Southern Ocean. Sea spray aerosol is an important contributor to total aerosol radiative forcing and poorly constrained in climate models; hence how it may change in a warming world is not well understood. The analysis is thorough and gives a comprehensive picture of sea spray aerosol properties (which are assumed to be the dominant source of marine aerosol during austral winter).

We thank the referee for the encouraging words and constructive suggestions. Please find below point by point reply to your comments.

Where I would like to see improvement (which I think will also enhance the impact of the paper), is in thinking about how the results are meaningful. They are nicely summarised in the conclusions, but a discussion/synthesis regarding importance and outlook is missing. The authors discuss that climate models represent SSA poorly in the introduction (lines 90-109) – can the work be linked back to that discussion, for example? Do the results shed new insight into what climate models are doing wrong and how SSA parametrizations could be improved? The end of the introduction (lines 121-127) could also be modified to address the significance of the objectives.

The following text will be added to the end of the 'Conclusions' section as a synthesis of the work carried out here and the outlook.

"This study elaborates on the relative importance of RH on the optical properties of sea salt over the Southern oceans during SH winter months when other natural emissions and long range transport of anthropogenic emissions are negligible. Although a handful of in situ measurements have been carried out using ship cruises and aircrafts, this is the first study of its kind that has attempted a detailed mapping of the optical properties of sea salt in its natural environment on the basis of the three fundamental parameters that the emissions depend upon - RH, wind speed and sea surface temperature and their vertical distribution purely from satellites. The generation of sea salt aerosols in the models are described in terms of source functions that depend on either one or more of the factors mentioned above for varying size ranges and these are derived from lab or ship/aircraft measurements. These source functions are adapted depending on the model to get the emissions in the correct order so as to accurately estimate the total aerosol optical depth (for example, some models introduce a temperature dependency or alter the power factor so as to increase/decrease the wind speed dependency or include the super micron size particles in the calculations). Also, due to the high hygroscopicity of sea salt, the parameterization schemes simulate the emissions at 80% RH (Monahan, et al., 1986). To introduce the variability of the water uptake by dry sea salt particles, recent models use hygroscopic growth factors at varying RH that are prescribed (Chin et al., 2002) or that are based on empirical equations or are explicitly calculated (Ghan et al., 2001; Vignati et al., 2004). The dependency of the aerosol optical properties on RH is of profound importance for climate forcing estimates, particularly, in the case of sea salt as they can co-exist in both phases at the same time between 50% and 70% RH. And this dependency changes when sea salt particles are contaminated by other chemical species. This study here shows the variability in the sea salt aerosol optical properties on a range of observed relative humidities, water temperatures and wind speeds over the Southern Oceans. In this study, we also generate fitting functions for aerosol extinction and backscatter based on wind speed and relative humidity for naturally occurring sea salt which can in turn, help improve the source functions used in the sea salt parameterization schemes. Further insights are needed to map the behaviour of the optical properties of sea salt when they are coated with other naturally occurring chemical components as is in the case of the rest of the oceans. "

Methods – I was interested to know how often cloud-free pixels were found in the data – after fulfilling all the selection criteria, what fraction of data are included in the analysis?

The total cloud cover over the Southern Ocean ranges from 75%-85% depending on the winter month and region. However not all clouds are optically thick and the CALIOP lidar is able to see through the thin clouds providing retrievals in nearly 40-50% of the cases. Here we have therefore used all-sky retrievals. The criterion of using only those retrievals that successfully converge without the change in initial lidar ratio (i.e. Extinction quality flag == 0) results in retaining nearly all aerosol retrievals. The cases when the lidar ratio was changed were less than 1% in total. Among all the retrievals available to compute mean climatological features, the scenarios representing the enhanced cyclonic and anticyclonic conditions represented about 27% and 17% of those cases respectively. We have added this text in the revised manuscript.

Lines 158: A reference is needed to justify the use of the 700 hPa GPH. Eg https://agupubs.onlinelibrary.wiley.com/doi/full/10.1029/2005GL022419

or https://journals.ametsoc.org/view/journals/clim/18/5/jcli-3284.1.xml

These references are added in the revised manuscript.

In the paper's present form, the readers are supposed to take it as read that the 25th and 75th percentile thresholds of GPH correspond to ascending and descending air masses; further explanation/references are needed here, which links to my next point.

We have now clarified in the manuscript that 25th/75th percentile corresponds to ascending/descending airmasses. The figures and text are adapted to follow only one set of terminology - ascending and descending airmasses.

Line 184. I follow the logic here, but because cyclonic conditions are defined as less than the 25th percentile in GPH, it's slightly confusing that they are also representative of the mean state. A plot of the GPH distribution (either in the Methods or Supplementary info) would help.

Southern Oceans are predominantly a region of intense extratropical cyclonic activity. More than half of the winter cyclones have a structure that extends through to the lower atmospheric levels (Simmonds and Keay 2000a; Hougthon et al. 2001; Lim and Simmonds 2002; Lim and Simmonds, 2007). Hence, it is not surprising that the 25th percentile of GPH that corresponds to cyclonic conditions is representative of the mean state.

A plot of the GPH distribution is added to the revised manuscript.

Figures: the use of the red/green colormap cannot be interpreted by those with color blindness; see e.g. https://www.scientificamerican.com/article/end-of-the-rainbow-new-map-scale-is-more-readable-by-people-who-are-color-blind/

A different colormap is used and all the figures are modified accordingly.

Instead of labelling the figures with 'P25' and 'P75' in the legend, using the labels 'ascending' and 'descending' (or similar) would make interpretation easier.

The labels are re-written in the revised manuscript.

**Minor comments:**

L45. Aerosol radiative forcing has been updated toward more negative values, and is estimated to be between -2 to -0.4 W m-2 (90% likelihood). https://agupubs.onlinelibrary.wiley.com/doi/full/10.1029/2019RG000660

Replaced the sentence by: "A recent study reports the global mean total effective aerosol radiative forcing to be in the range of -2.0 to -0.35 W/m2 when constrained by observations, with estimates of aerosol-cloud interactions in the range of -2.65 to -0.07 W/m2 (Bellouin et al., 2020)"

L203-204. Definitions of efflorescence and deliquescence are given here; it would be helpful to include these in the introduction also, around lines 59-65.

The definitions are added in the introduction.

L 80-81. Unclear – at what wind speeds are 26% of the accumulation mode particles sea salt?

This information is added and the sentence is rephrased as: "It was observed that 26% of the accumulation mode particles below 200 m was sea salt and this fraction increases with wind speed by 11% for wind speeds less than 4 m/s and respectively, 20% and 30% at 4-8 m/s and 8-12 m/s."

L146-147. I assume that avoiding the sea-ice zone further reduces the risk of contamination from DMS-derived sulfate aerosol (which should have low concentrations during austral winter anyway).

Yes, that is one of the reasons to avoid the sea-ice zones. The other reason is to avoid potential contamination and misclassification due to wind-blown snow/ice crystals in CALIOP-CALIPSO retrievals. This is already mentioned in the manuscript under Section 2.

Line 198. Further explanation needed – why is it remarkable that linear depolarization is not sensitive to varying winds?

We find it remarkable from two perspectives: First, as the wind speed increases, one would expect drying of sea salt particles and hence, higher linear depolarization. This is especially true above the boundary layer where not only the winds are stronger but the humidity is lower. Second, models for optical properties of marine aerosols show that the linear depolarization ratio is often increasing with the volume-equivalent particle radius - see, e.g.

https://agupubs.onlinelibrary.wiley.com/doi/full/10.1029/2020JD033674

Correspondingly, one would expect a generally increasing trend of the depolarization ratio with wind speed.

We have clarified these points in the revised manuscript.

**References:**

Chin, M., Ginoux, P., Kinne, S., Torres, O., Holben, B. N., Duncan, B. N., Martin, R. V., Logan, J. A., Higurashi, A., and Nakajima, T.: Tropospheric aerosol optical thickness from the GOCART model and comparisons with satellite and sun photometer
measurements, J. Atmos. Sci., 59, 461–483, 2002.

Ghan, S. J., Laulainen, N., Easter, R. C., Wagener, R., Nemesure, S., Chapman, E., and Leung, Y. Z. R.:
Evaluation of aerosol direct
radiative forcing in MIRAGE, J. Geophys. Res., 106, 5295–5316, 2001.

Houghton, J. T., Ding, Y., Griggs, D. J., Noguer, M., van der Linden, P. J., Dai, X., Maskell, K., and Johnson, C. A.: Climate Change 2001: The Scientific Basis. Cambridge University Press, 881pp, 2001.

Lim, E.-P. and Simmonds, I.: Southern Hemisphere Winter Extratropical Cyclone Characteristics and Vertical Organization Observed with the ERA-40 Data in 1979–2001, J. Climate, pp. 2675–2690, https://doi.org/10.1175/JCLI4135.1, 2007.

Lim, E.-P. and Simmonds, I.: Explosive cyclone development in the Southern Hemisphere and a comparison with Northern Hemisphere events., Mon. Wea. Rev., p. 2188–2209,

https://doi.org/10.1175/1520-0493(2002)130<2188:ECDITS>2.0.C, 2002.

Monahan, E. G., Spiel, D. E., and Davidson, K. L.: A model of marine aerosol generation via whitecaps and wave disruption, in Oceanic Whitecaps, edited by: Monahan, E. and Niocaill, G. M., 167–174, D. Reidel, Norwell, Mass., 1986.

Simmonds, I. and Keay, K.: Variability of Southern Hemisphere extratropical cyclone behavior, J. Climate, p. 550–561, https://doi.org/10.1175/1520-0442(2000)013<0550:VOSHEC>2.0.CO;2, 2000

Vignati, E., Wilson, J., and Stier, P.: M7: a size resolved aerosol mixture module for the use in global aerosol models, J. Geophys.
Res., 109, D22202, doi:10.1029/2003JD004485, 2004.

**Response to Reviewer #2**

I have now read the manuscript of Thomas et al. which concerns (optical) properties of the Southern Ocean marine aerosols. My overall recommendation is major revision. Indeed, the present manuscript suffers some flaws that need to be addressed. I also have some requirements and points the authors must clarify. Below are some comments.

We thank the referee for the constructive suggestions. Please find below point by point reply to your comments.

- The authors should present a geographic map to locate the investigated area(s).

This is added in the revised manuscript.

- An important reference (to be quoted) about marine aerosols in the Indian Southern Ocean is Mallet et al. (2018) "Marine aerosol distribution and variability over the pristine Southern Indian Ocean" . Atm. Env. 182.

Thanks for this very relevant reference, which is now added in the revised manuscript.

l. 41 : interference → influence

Corrected.

l. 104 : Indicate what is $u\_10^{3.5}$. It is clear but explicit indication would help some readers

The sentence is now rephrased.

l. 143 "their contribution was little and did not make much difference to" : This must be quantified

The range is now added

"The extinction quality flag of 0 ensures that the retrieval converges successfully and there is no need to change the initial lidar ratio. We also investigated additional cases with extinction quality flags of 1 and 2, wherein the lidar ratio was adjusted, but their contribution was little (0.25% and 0.6% respectively) and did not make much difference to the overall statistics presented here."

l. 146: "The profiles over the sea-ice and within 50 km from the ice-edge are removed to avoid potential contamination and misclassification due to the wind-blown snow/ice crystals." : This sentence is unclear and must be rephrased.

The sentence is rephrased as "In order to avoid potential contamination and aerosol misclassification in CALIOP-CALIPSO aerosol retrievals due to wind-blown snow/ice crystals, the data profiles over the sea ice and up to 50 km from the ice -edge are not included in this analysis."

l. 167: How did the authors obtain the backscatter coefficient? And, later, the extinction coefficient? Data? Model? Is it a coefficient or a cross section? Particles are supposed to be spherical in the calculation?

The section 2, 'Data and Methods' has been reorganized and revised to address the questions raised here. We make use of the vertical profiles of the aerosol properties such as extinction and backscattering coefficients and linear depolarization ratio, all retrieved from the CALIOP sensor data (flying onboard CALIPSO satellite). Please note that this study is purely based on these satellite observations and no modelling is performed.

The original Algorithm Theoretical Basis Documents on how these retrievals were performed can be found here:

https://www-calipso.larc.nasa.gov/resources/pdfs/PC-SCI-202.Part1_v2-Overview.pdf
https://www-calipso.larc.nasa.gov/resources/pdfs/PC-SCI-202_Part2_rev1x01.pdf
https://www-calipso.larc.nasa.gov/resources/pdfs/PC-SCI-202_Part4_v1.0.pdf

We have used aerosol retrievals of backscatter, extinction, and depolarization provided in the most recent Version 4 data products. There is a special issue in the EGU journal of Atmospheric Measurements and Techniques dedicated to the Version 4 products.

https://amt.copernicus.org/articles/special_issue903.html
https://amt.copernicus.org/articles/11/5701/2018/

We have clarified this in the revised manuscript.

P25 and P75 must be explicitly specified. 25%ile and 75%ile are not standard notations.
They have been re-written and referred to as air masses that are 'ascending' or 'descending'.

l.170: Is it the optical index or the refractive index (real part of the optical index)?
It is the *complex refractive index*, which, we believe, it the standard term used throughout most of the aerosol-optics standard literature, e.g. in the monographs by van de Hulst (Light Scattering by Small Particles, Dover, New York, 1981), Bohren and Huffman (Absorption and Scattering by Light by Small Particles, Wiley, Weinheim, 1983), and Mishchenko, Travis, and Lacis (Scattering, Absorption, and Emission of Light by Small Particles, Cambridge University Press, Cambridge, 2002). We have added "complex" to "refractive index" in the revised manuscript to avoid any confusion.

l. 175 and around: The considerations given must be quantified.
The sentence is rephrased as "An increase in both the number concentration and the mean size of sea salt aerosols contribute to an increase in aerosol backscatter values ranging from 0.005-0.012 $km^{-1}sr^{-1}$ at wind speeds below 14 m/s to above 0.018 $km^{-1}sr^{-1}$ (reaching as high as 0.025 $km^{-1}sr^{-1}$ ) at wind speeds stronger than 14 m/s." in the revised manuscript.

Fig.1 and others: It is quite strange to have a decreasing axis of RH. I assume it is because absolute humidity is higher close to the sea. It is not necessary the same for RH since it depends on temperature.
Please note that although RH is on Y-Axis, it has nothing to do with the height. We understand that the absolute humidity is higher close to the sea and not the RH. However, our area of interest is over the Southern Oceans where strong winds are encountered making the boundary layer well mixed. The chances of temperature inversions occurring under such conditions are very rare. For this reason we anticipate a decrease in RH with height as is in the case of absolute humidity. One other reason to use the RH instead of absolute humidity is because all the parameterization schemes of sea salt are based on RH.

l.190-195: The authors talk about the aerosol extinction, so do they really include absorption in their discussion? Sea salt are not very absorbing. They say that 'similar conclusions can be derived for the aerosol extinction'. This must be detailed a bit more. Also, what is the relation with RH and aerosol size?
We do not understand the reviewer's question about the inclusion of absorption, nor the concern that marine aerosols are only weakly absorbing. First, as a side note, we mention that modelling studies (Kanngießer and Kahnert, JGR 2021) show that even small to moderate values of the imaginary part of the complex refractive index, e.g. 0.002 (as measured by Shettle and Fenn, 1979) and 0.059 (as measured by Hänel, 1976) can have quite an impact on depolarisation and backscattering by marine aerosol. Second, and more importantly, we do not quite understand why the reviewer would object to considering extinction in the absence of absorption (unless we have misunderstood this comment). Since the extinction coefficient is the sum of the scattering coefficient and the absorption coefficient, this quantity will always be larger than zero in the presence of aerosols, even for non-absorbing aerosols. In the absence of absorption, the extinction coefficient is equal to the scattering coefficient. We can only conjecture that the reviewer wonders whether extinction and backscattering coefficients, in such a case, will contain redundant information. However, this is most certainly not the case.

In our original manuscript, assuming that our readers understand these concepts, we did not explain the difference between the *scattering* coefficient and the *backscattering* coefficient. The former is an integral radiative property that depends on the total scattering cross section (i.e., accounting for scattering in *all directions*), the latter is a differential radiative property that depends on the differential scattering cross section in the *backscattering direction* (summed over incident and averaged over scattered polarisation states). So, even in the absence of absorption, these two quantities are distinct (even though the specific CALIOP retrieval of the extinction coefficient has to rely on a priori information). Both quantities are CALIOP data products that are available regardless of whether the aerosols absorb or not. Third, the CALIOP data we use in this study do not provide us with any information on whether or not the aerosols are absorbing. Thus our use of the extinction-coefficient data product does neither entail nor depend on any assumptions on the absorption properties of the aerosols we study.

To make our presentation more self-contained, we now subdivided section 2 into two subsections. The first subsection is called "Optical properties of aerosol particles"; it provides definitions of the extinction coefficient, backscattering coefficient, and the linear backscattering depolarization ratio. It also explains the relevance of these quantities for the purpose of our study. The whole subsection consists of four new paragraphs, which we hope will help readers unfamiliar with aerosol optics to better follow the main body of the paper. The second subsection is labelled "Data" and explains the satellite products we are employing in our analysis.

The sentence "similar conclusions can be derived for the aerosol extinction" is deleted and is now explained clearly in the revised manuscript.

l. 212: water absorption → water uptake
Corrected.

Legend of Fig.4: symbol for steradian is sr and not Sr
Corrected.

Section 3.3: The authors use a vertical distribution. They must either include an altitude axis z (km or m) to help readers understanding or convert pressure into altitude.
The vertical levels are given in pressure coordinates as they are widely used in the modelling community.

It is not surprising that total backscattering increases with RH and p: RH provides water vapour and high pressure and low temperature (low thermal agitation) favor water uptake. Figure 6 is not at all unexpected. So, do the results obtained really original? This is a serious flaw of the paper: the authors present scatterplots and make only general comments on what they see on their figures. Also, it would be a clear advance in the field to propose a parameterization of the relations between RH and backscattering and extinction (and/or absorption). Moreover, since RH contains both absolute humidity (AH) and T, the same study using AH would be useful. Such considerations are important then for modeling. We appreciate this comment made by the reviewer. The results shown are novel because, for the first time, we provide quantitative analysis of the dependence of sea-salt aerosol properties on the meteorological conditions and that too using purely independent satellite based observations. We also cover the entire Southern Ocean. All the previous studies are based on lab measurements or aircraft/ship observations that followed a specific track. With the recent advancements in satellites, we can now retrieve the aerosol optical properties over the Southern oceans with high confidence even during the winter season where other observations are not available. It is indeed a good idea to propose a relationship between the different aerosol optical properties based on the basic meteorological parameters. In the revised manuscript we have fitted dependencies of extinction as a function of wind speed and humidity and have provided these equations which could be used for improving the parameterizations by the modelling community.  An example of a third degree polynomial fit estimating extinction at 532 nm as a function of wind speed is shown below.

[Figure]

EXT = 0.000043 * WS³ − 0.000492 * WS² + 0.000205 * WS + 0.087285

Finally, since the authors recall in their introduction that marine aerosols are important for climate change, they should estimate radiative forcing due to water uptake by marine aerosol (it is not enough for me to focus only (qualitatively) on data about RH and optical properties as done here).

It is not straightforward to estimate the radiative forcing depending on the water uptake based on the parameters we have here. It would require a lot more assumptions and additional parameters, at least the single-scattering albedo and the asymmetry parameter of the aerosol, and we think it is beyond the scope of this study to estimate this. Moreover, the main focus of this study is to show the dependency of backscattering, extinction, and depolarisation of marine aerosol on meteorological parameters such as RH, wind speed and SST on which the majority of sea salt parameterization schemes are based.

l.278: cubes → polyhedrons

We would prefer to keep "cubes", because we want to express here that cubes are a canonical reference shape for sea-salt aerosol in much the same way as hexagons are a reference shape for ice crystals. Note that we do not claim that sea-salt particles are cubes. We merely say that they "have shapes similar to cubes". Pure sodium chloride does have fcc crystal structure, so that halite crystals grown under ideal conditions have, indeed, the shape of perfect cubes, just like ice crystals under certain ideal conditions have hexagonal shape. These canonical reference shapes are often employed as a starting point by modellers to devise more realistic model particles, such as irregular ice aggregates assembled out of hexagons, or deformed cubes as a model for non-ideal sea salt. We do not want to use the term "polyhedron", as it is extremely general and generic and does not really say much at all.

l.285: depolarization is interesting to discriminate dry sea salts from wet ones. Can the authors use their data of depolarization to perform such a discrimination? It is essential since both kind of aerosols do not have the same radiative (direct/indirect) effects. This would be an added value to the study presented.

This is an extremely challenging problem, which we are presently tackling from two sides. On the optics-modelling side, we are developing aerosol-optics models for sea-salt. For instance, in Kanngießer and Kahnert (JGR 2021) we study different modelling approaches for dry sea salt. As the reviewer correctly notes, depolarisation will provide very valuable information, as it allows us to obtain information about the presence of dry, nonspherical particles. However, we believe that it should be possible to obtain even more detailed information about the amount of water uptake. But to retrieve this information, we first need to develop a quantitative understanding of how the depolarisation signal depends on the amount of water uptake. We recently performed another modelling study (Kanngießer and Kahnert, Opt. Express 2021), which focuses on the effect of water uptake on depolarisation. This modelling work helps us to understand the relation between physical properties (size, water uptake, morphology), and optical properties. On the observation side, we investigate the relation between meteorological conditions and optical properties. This

is what we do in this manuscript. The critical link between the modelling and the observation approach are the parameterisation schemes employed in aerosol transport and Earth-system climate models, which connect meteorological conditions to physical particle properties. Our final goal is to put all of these three puzzle pieces together, so that we can cover the whole chain from meteorology to physical properties to optical properties. In this effort, the observations will help us to evaluate the parameterisation schemes employed in aerosol transport models. The present study is meant to provide the observational benchmarks, at least for the Southern Ocean under wintertime conditions. We are not aware of any other study that provides anything comparable based on such a comprehensive body of observational data.

In the conclusion, joint histograms are mentioned. I did not see any histogram in the manuscript.
We believe the Figs 1-8 show joint histograms, because they show occurrences as a function of two variables jointly and simultaneously.

**References:**

van de Hulst, H. C.: Light scattering by small particles. Published by New York (John Wiley and Sons), London (Chapman and Hall), 1957. Pp. xiii, 470; 103 Figs.; 46 Tables. 96s. Q.J.R. Meteorol. Soc., 84: 198-199, 1958. https://doi.org/10.1002/qj.49708436025, 1958.

Bohren, C. F. and Huffman, D. R.: Absorption and Scattering of Light by Small Particles, by Craig F. Bohren, Donald R. Huffman, pp. 544. ISBN 0-471-29340-7. Wiley-VCH , 544, 1998.

Mishchenko, M. I., Travis,L. D. and Lacis, A. A.: Scattering, Absorption, and Emission of Light by Small Particles. Cambridge University Press, 2002.

Shettle, E. and Fenn, R.: Models for the Aerosols of the Lower Atmosphere and the Effects of Humidity Variations on their Optical Properties. Environ. Res.. 94, 1979.

Hänel, G.: The Properties of Atmospheric Aerosol Particles as Functions of Relative Humidity at Thermodynamic Equilibrium with Surrounding Moist Air. Advances in Geophysics, 19, 73-188. http://dx.doi.org/10.1016/S0065-2687(08)60142-9, 1976.

Kahnert, M. & Kanngiesser, F.: Aerosol optics model for black carbon applicable to remote sensing, chemical data assimilation, and climate modelling. Optics Express, 29(7), 10639-10658, 2021.